# Use of Undigested NDF for Estimation of Diet Digestibility in Growing Pigs

**DOI:** 10.3390/ani10112007

**Published:** 2020-10-31

**Authors:** Marco Battelli, Luca Rapetti, Andrea Rota Graziosi, Stefania Colombini, Gianni Matteo Crovetto, Gianluca Galassi

**Affiliations:** Dipartimento di Scienze Agrarie e Ambientali—Produzione, Territorio, Agroenergia, Università degli Studi di Milano, via Celoria 2, 20133 Milano, Italy; marco.battelli@unimi.it (M.B.); andrea.rota@unimi.it (A.R.G.); stefania.colombini@unimi.it (S.C.); matteo.crovetto@unimi.it (G.M.C.); gianluca.galassi@unimi.it (G.G.)

**Keywords:** uNDF, internal marker, diet digestibility, pigs

## Abstract

**Simple Summary:**

Knowledge of diet digestibility in animals is essential to reduce feed costs and the amount of undigested nutrients excreted in the manure. Diet digestibility can be evaluated via in vivo, in situ (i.e., in the rumen of cannulated animals), and in vitro methods. The in vivo by total faecal collection methods and the in situ methods are expensive and ethically questionable. The in vitro methods are cheaper but also less accurate. This work aimed to verify whether the in vivo method with undigested neutral detergent fibre (uNDF), which is commonly used in ruminants as an internal marker, could be adapted toward growing pigs. Dry matter, organic matter, and neutral detergent fibre digestibilities estimated with the uNDF were compared with in vivo values determined by total faecal collection in a previous study. The effects of pre-treating samples with the neutral detergent solution and adding α-amylase were also tested to improve the repeatability and accuracy of the results. It was concluded that the estimation of diet digestibility with pre-treated uNDF as an internal marker in growing pigs could be an alternative to the total faecal collection method.

**Abstract:**

Undigested neutral detergent fibre (uNDF) is commonly used as an internal marker for the estimation of diet digestibility in ruminants. This work aimed to verify (i) whether the in vivo method with uNDF could be used to evaluate diet digestibility in growing pigs, and (ii) whether pre-treating the samples with neutral detergent solution (NDS) and α-amylase improves the accuracy of the estimates. Samples from a previously published work of two diets with known in vivo digestibility values estimated by the total faecal collection method and 16 individual samples of faeces were used. For each sample, four Ankom F57 bags were weighed. Before the incubation, two F57 bags were pre-treated with NDS and α-amylase. All the samples were incubated for 240 h in the Ankom Daisy^II^ incubator and then analysed for their uNDF contents. Dry matter, organic matter, and neutral detergent fibre digestibilities were estimated using the uNDF contents, and the results were compared with those of the former study. The digestibility values obtained using the uNDF method with pre-treatment were not statistically different from those determined with the total faecal collection. On the contrary, the uNDF method without the pre-treatment could not satisfactorily predict the digestibilities of pig diets.

## 1. Introduction

The highest expenditure in pig farming is attributed to feed and piglet costs, which can together reach 90% of the total production cost [1]. In order to reduce both economic and environmental costs, it is necessary to optimise nutrient efficiency [2]. Precision feeding techniques can increase the economic and environmental sustainability of pig farming systems. As reported by Andretta et al. [3], precision feeding could reduce nitrogen excretion by 38% and feeding costs by 10% compared with conventional phase-feeding systems.

Digestion is a complex process influenced by both the characteristics of the animal (such as species and breed, genotype, sex, physiological stage, and body weight) and feeds (e.g., chemical composition and processing treatments) [2]. However, in order to formulate a balanced diet, it is also necessary to understand the interaction between the animal and its feed, as the latter affects the total tract digestibility.

Feed digestibility in pigs can be measured directly by the total collection of animal faeces. These in vivo studies are expensive both economically and in terms of time [4]. Moreover, they are stressful for the animals as they must be isolated and caged. For these reasons, feed digestibility is frequently estimated via prediction equations based on the chemical composition of the feed [2,4]. In this regard, near infrared spectroscopy technology could be used to predict diet digestibility, starting with faecal analyses. However, additional studies are required for appropriate calibration of the instruments [4]. In addition, feed digestibility may be measured indirectly using marker substances [5,6]. The reliability of the chosen marker is assessed in terms of the recovery of the marker itself. This recovery is expressed as the quantity of the marker recovered from the total collection of faeces or the proportion of the total marker amount consumed. Theoretically, the value of the marker recovery should be constant and close to 1 [5,6]. Frequently employed inert markers include minerals such as chromic oxide (Cr_2_O_3_) and titanium dioxide (TiO_2_) [5,6], but internal markers such as acid-insoluble ash [5] and lignin [6] can also be used. Undigested neutral detergent fibre (uNDF) is a commonly used internal marker to estimate in vivo diet digestibility in ruminants [7,8,9,10]. uNDF is the laboratory estimation of the undigestible neutral detergent fibre, typically after 240 h of in vitro incubation.

The filter bag technique with the Daisy^II^ incubator (ANKOM Technology Corp., Fairport, NY, USA) is an in vitro protocol used for the measurement of uNDF. This method simplifies the measurements of uNDF and in vitro dry matter digestibility by eliminating the need to filter the samples after incubation [11]. Moreover, the Daisy^II^ incubator allows testing of several samples simultaneously [12]. This technique is commonly employed for ruminants and has also been successfully used for horses and donkeys [13,14].

Several studies have verified the precision and accuracy of the Daisy^II^ protocol, albeit with different results. Tagliapietra et al. [15] analysed in situ true dry matter digestibility (DMD) with either traditional nylon bags or synthetic filter bags, and in vitro, with either conventional bottles or the Daisy^II^ incubation technique. They found that the DMD values obtained with the Daisy^II^ incubator were lower compared to those observed with the in situ nylon bags and in vitro conventional bottles. Moreover, the low repeatability of this system was overcome by increasing the number of replicates: using three filter bags provided approximately the same standard error as the mean of two conventional bottles. Tagliapietra and colleagues also observed direct proportionality between the DMD values of different in situ and in vitro techniques.

The starch present in the feed is a source of variability [16,17]. On the one hand, due to their small size (<50 µm), the starch granules could leak out from the filter bag, causing an overestimation of digestibility [16]. On the other hand, starch is digested at a faster rate, thus causing more rapid gas production compared to fibre. In our preliminary works (Battelli et al., unpublished data), it was observed that the gas produced in the starch-rich samples inflated the filter bags. This fact might reduce the filtering capacity, leading to an underestimation of fibre digestibility. These problems could occur with pig diets, commonly rich in starch and with low fibre content. However, pre-treating the samples with a neutral detergent solution (NDS) and α-amylase may be a solution to this problem, as doing so would hydrolyse the starch and the other non-fibre carbohydrates.

To the best of our knowledge, the uNDF technique has never been used to estimate diet digestibility in pigs. The aims of this work were to (i) verify whether the in vivo technique with the uNDF marker can be used for estimating the digestibilities of pig diets and (ii) determine whether this technique is more accurate with or without pre-treatment with NDS and α-amylase. In order to achieve the goals of the work, samples of pig diets and faeces were analysed, and in vivo total faecal collection diet digestibility values obtained from a previously published study [18] were used as a reference.

## 2. Materials and Methods

The study was conducted at Cascina Baciocca, the experimental farm of the University of Milan. Authorisation for the study (authorisation no. 904/2016-PR) was obtained from the Ministry of Health.

In a previous work [18], 16 pigs with an average body weight of 128 ± 10 kg were divided into 2 groups and fed 2 different diets formulated to meet the needs estimated by the National Research Council (NRC) Nutrient Requirements of Swine [19]. The first diet was a conventional diet (CONV) containing cereal meals (corn, barley, and wheat; 691 g/kg as fed), soybean meal (91.7 g/kg as fed), wheat bran (80 g/kg as fed), wheat middling (60 g/kg as fed), minerals, and supplements. The second diet was a low-protein (LP) diet without soybean meal, with more cereal meal, and supplemented with crystalline amino acids. The results of the analyses of the CONV and LP diets, respectively, were as follows (g/kg as fed): dry matter—882, 883; ash—44, 40; ether extract—40, 38; crude protein—132, 104; neutral detergent fibre—121, 129; acid detergent fibre—38, 39; and starch—443, 486. All the pigs were housed individually in metabolic cages for 14 d—7 for the adaptation and 7 for the faeces collection.

### 2.1. In Vitro uNDF Determination

The in vitro incubations were conducted on both diets and the 16 individual faecal samples collected during the in vivo digestibility trial [18]. All the samples (diets and faeces) were dried at 60 °C for 48 h in a forced-air oven and ground to pass a 1-mm Fritsch mill (Fritsch Pulverisette, Idar-Oberstein, Germany). The dry matter (DM) and ash contents were, respectively, determined by heating at 105 °C for 24 h [20] and incineration at 550 °C for 5 h [20]. The NDF content was calculated using an Ankom^200^ Fiber Analyzer (Ankom Technology Corporation, Fairport, NY, USA) following the procedure reported by Mertens [21]. For each sample of the two diets, eight Ankom F57 bags were weighed (Ankom Technology, Macedon, NY, USA) (each bag contained 0.500 g of sample). For each of the 16 faecal samples, 4 F57 bags were prepared in the same way. Eight blank Ankom bags (i.e., bags without a sample) were also weighed (two for each of the four jars of the in vitro incubation system described below). Before the incubation, four bags of each diet and two bags of each faecal sample were subjected to pre-treatment consisting of washing with NDS and α-amylase [21].

The incubation was conducted using the Daisy^II^ incubator (Ankom Technology, Macedon, NY, USA) at 39 ± 0.5 °C. In addition to the two blanks, each of the four jars of the apparatus contained two Ankom F57 bags of each diet (one pre-treated and the other untreated) and one faecal sample of each animal.

Two rumen-cannulated cows (non-lactating Holstein–Friesian) were used as donors of ruminal fluid. The cows were fed with 70:30 forage:concentrate diet twice a day. The rumen fluid was collected before the morning meal and was immediately strained through four layers of cheesecloth into a pre-warmed (39 °C) flask, flushed with CO_2_, and mixed with the buffer solution prepared as per the Ankom protocol [22]. The inocula were renewed at 120 h (i.e., the contents of the jars were replaced with a new inoculum). The duration of the incubation was 240 h, and the samples were analysed for their uNDF contents at the end of the incubation [21].

### 2.2. Digestibility Determination with uNDF as the Internal Marker

The in vivo apparent total-tract nutrient digestibilities were calculated by using the uNDF content as the internal marker for both the diets and the faeces, as described by Schalla et al. [23]. The following equation was used:
Apparent nutrient digestibility (%) = 100 − [100 × (diet uNDF/faecal uNDF) ×(faecal nutrient content/diet nutrient content)](1)

All values as % of DM unless otherwise noticed.

### 2.3. Statistical Analysis

The digestibility data obtained by the two methods (in vivo digestibility with total faecal collection vs. in vivo digestibility with uNDF as the internal marker) were analysed and compared using the GLM procedure of SAS (version 9.4, SAS Institute Inc., Cary, NC, USA) while considering the main effects of diet and pre-treatment as well as their interactions.

## 3. Results

The results for the digestibilities obtained using the uNDF method as the internal marker with and without the NDS and α-amylase pre-treatment and those from the previous in vivo study [18] are shown in Table 1.

When the uNDF method was used in tandem with the NDS and α-amylase pre-treatment, the digestibility results were not significantly different (*p* > 0.05) from those provided by the total faecal collection method. Moreover, the diet did not influence the results of the DM and organic matter (OM) digestibilities with the two methods. Instead, when the pre-treatment was not performed, the method with the uNDF as the internal marker returned digestibility values statistically lower (*p* < 0.05) than those obtained with the total faecal collection method. Figure 1 shows the differences in OM digestibility between the two uNDF methods and the total faecal collection method for each pig.

In addition, it was found that the uNDF method without pre-treatment determined significant differences between the two diets for DM, OM, and NDF digestibilities (*p* < 0.05); all the values estimated for the CONV diet without pre-treatment in the uNDF method were statistically higher than those for the LP diet, unlike the findings from the total faecal collection and pre-treated uNDF methods.

## 4. Discussion

The digestibility of feed is a fundamental parameter that must be known in order to formulate diets that meet the needs of the animals as well as minimise waste. The knowledge of diet digestibility allows us to formulate balanced diets while simultaneously reducing food costs and the environmental impact of livestock farming. This study aimed to evaluate the applicability of the in vivo technique with the uNDF as internal marker toward estimating the digestibilities of pig diets and to compare its results with the in vivo by total faecal collection digestibility values obtained in a previous work [18].

Galassi and colleagues [18] reported no difference between the two diets and the two genetic groups of pigs used in their study on the in vivo apparent digestibilities of DM, OM, and NDF. Furthermore, Galassi et al. [18] tested the interaction between diet and genetic type but observed no positive findings. Therefore, for the present study, the data of the two genetic groups were considered together.

The incubation for the uNDF determination was conducted with the Daisy^II^ incubator. The Daisy^II^ incubator is an excellent tool to analyse different feedstuffs, improve the precision and reproducibility of an assay, and reduce the time and costs of analysis [24]. The Daisy^II^ incubator is commonly used to estimate in vitro diet digestibility in ruminants, and satisfactory results have been reported for cows, sheep, and goats. The use of the Daisy^II^ incubator to evaluate diet digestibility for ruminants does not require pre-treatment with NDS and α-amylase due to the low starch contents of ruminant diets. To the best of our knowledge, few researchers have used the Daisy^II^ incubator to study pig diets. Pig diets are characterised by a high starch concentration, which can cause some technical drawbacks when using Ankom bags.

uNDF is used in ruminants as an internal marker to predict digestibility. For example, Lopes et al. [25] used uNDF as an internal marker at 120 h to estimate in vivo nutrient digestibility. Lund et al. [26] compared the in situ uNDF and chromic oxide as markers at 504 h to estimate NDF digestibility in dairy cows. They found that using uNDF was preferable. Similarly, Morris et al. [27] compared uNDF and acid-insoluble ash as internal markers at 288 h against total faecal collection in order to estimate DM, OM, and nitrogen digestibilities in dairy cows. They found that uNDF was a better faecal marker than acid-insoluble ash, and given the variation in the faecal marker concentration, they recommended a spot sampling frequency of at least six times a day. De Carvalho et al. [28] compared undigestible DM, in situ uNDF at 240 h, and undigestible acid detergent fibre as internal markers in sheep and goats. They reported that the undigestible DM and uNDF exhibited full faecal recovery for both sheep and goats, thus pointing to their suitability for use in digestion trials.

Three analytical in vitro methods were developed to estimate diet digestibility for swine [29]. The first is a one-step method that mimics gastric digestion. The second is a two-step method, which simulates small intestine digestion in addition to gastric digestion. The last technique involves three steps; in addition to the previous ones, it simulates digestion in the large intestine. However, all these methods are more laborious and time consuming compared to the uNDF approach proposed in this study. Moreover, as these methods are totally in vitro, they are less accurate compared to the approach proposed in this study, which is an in vivo method with an internal marker.

The results of our study show that using the uNDF as an internal marker and pre-treating the samples with NDS and α-amylase provides DM and OM digestibility values that are statistically similar to those obtained with the total faecal collection method. Furthermore, the NDF digestibility values determined with the pre-treated uNDF method were not statistically different to those obtained with the total faecal collection method within the same diet. Conversely, without the pre-treatment, the DM, OM, and NDF digestibilities obtained with the uNDF marker were statistically lower than those determined with the total faecal collection method, as graphically shown in Figure 1 for OM. Presumably, the rapid starch fermentation altered the filter bag characteristics, causing an underestimation of digestibility. This reasoning is confirmed by the differences in the digestibilities between the diets, all of which are statistically significant; in particular, the NDF digestibility of the LP diet (the diet with the highest starch content) decreased by 32.2% compared to that of the CONV diet.

These results show that the method for determining digestibility with uNDF as the internal marker, which is already commonly used to predict digestibility in ruminants, can also be applied to pigs, provided that the diets and faecal samples are pre-treated with NDS and α-amylase. Indeed, after the pre-treatment with NDS and α-amylase was performed, the DM digestibility values obtained with the two methods (uNDF and total faecal collection) for both diets were extremely similar (all within a percentage point of difference). Even closer values of the four means of OM digestibility were obtained for both diets using these two methods. Slightly more pronounced differences, albeit within two percentage points of difference, were observed between these two methods for the two diets in terms of NDF digestibility. However, such differences were much smaller than those observed between the uNDF method without the pre-treatment with NDS and α-amylase and the total faecal collection.

The pre-treatment is necessary due to the high starch content of pig diets; in the absence of pre-treatment, this factor negatively affects incubation and leads to underestimations of digestibility values. Indeed, without the pre-treatment, the uNDF method provided significantly lower DM, OM, and NDF digestibility values than those obtained with the total faecal collection, especially for the LP diet, which is richer in starch.

Since in the present experiment the diets tested had about 12% NDF from good quality fibre feeds, we believe that this method (pre-treated uNDF) should be even more applicable to diets with higher fibre contents or less digestible fibre with a consequent higher uNDF concentration.

Finally, besides its promising results, this in vivo technique with the internal marker is cheaper, faster, more efficient, and more ethical than the in vivo method with total faecal collection, since animals are not required to be forced into cages for the sample collection.

## 5. Conclusions

The results of this study show that the proposed method with the marker, which is already commonly used in ruminants, is promising for evaluating the digestibilities of pig diets, provided that feeds and faeces samples are pre-treated with NDS and α-amylase. Validation on a broader dataset (in terms of diets and pig ages) is advisable in order to promote the use of this method for practical evaluations at commercial-scale farms.

## Figures and Tables

**Figure 1 animals-10-02007-f001:**
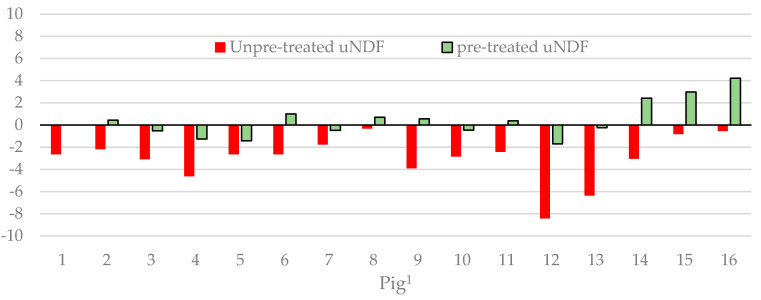
Differences in Organic Matter digestibility (percentage units) between the values predicted by the uNDF method (unpre-treated or pre-treated) and those determined by total faecal collection method for the 16 pigs. ^1^ Pig: from 1 to 8, pigs fed the conventional diet; from 9 to 16, pigs fed the low-protein diet.

**Table 1 animals-10-02007-t001:** Digestibility values (%) of the two diets measured with the undigested neutral detergent fibre (uNDF) method with and without the neutral detergent solution and α-amylase pre-treatment versus the total faecal collection method.

		CONV ^1^			LP ^2^		SEM		*p*-Value	
Item	Total Faecal Collection	Pre-treated uNDF	Unpre-treated uNDF	Total Faecal Collection	Pre-treated uNDF	Unpre-treated uNDF		D ^3^	M ^4^	D*M ^5^
DM ^6^	87.6 ^a^	87.5 ^a^	84.8 ^b^	86.9 ^a^	88.0 ^a^	82.8 ^c^	0.344	0.072	<0.001	0.037
OM ^7^	89.3 ^a^	89.1 ^a^	86.8 ^b^	89.5 ^a^	89.5 ^a^	85.0 ^c^	0.303	0.060	<0.001	0.039
NDF ^8^	57.7 ^a^	56.9 ^a b^	47.8 ^c^	54.8 ^a b^	52.9 ^b^	32.4 ^d^	1.174	<0.001	<0.001	<0.001

^1^ CONV: conventional diet; ^2^ LP: low-protein diet; ^3^ D: diet effect; ^4^ M: method effect; ^5^ D*M: interaction of the effects of diet and method; ^6^ DM: dry matter; ^7^ OM: organic matter; ^8^ NDF: neutral detergent fibre. ^a^, ^b^, ^c^, and ^d^: means in the same row with different superscripts are statistically different for *p* < 0.05.

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
