# Peer review of "Use of Undigested NDF for Estimation of Diet Digestibility in Growing Pigs"

_animals, 2020, doi:10.3390/ani10112007_

Round 1
Reviewer 1 Report
After corrections it is acceptable
Reviewer 2 Report
Authors significantly improved their manuscript. I propose to change the title from "Use of undigested NDF for estimation of diet digestibility in growing pigs" to "Use of undigested NDF for estimation of the organic matter digestibility in growing pigs" - which will be in line with the explanation Authors sent me in the response. The manuscript may be published in this form.
This manuscript is a resubmission of an earlier submission. The following is a list of the peer review reports and author responses from that submission.
Round 1
Reviewer 1 Report
Conclusions do not reflect the results presented in the table, because there was a significant interaction between a diet and the method. Apparently only pretreated uNDF could be used as a marker of digetibility.
Author Response
Dear reviewer, here in attachment you can find our reply to your comments.
Thank you.

Reviewer 2 Report
This manuscript describes an experiment that aims to i) verify whether the in vivo technique with the uNDF marker can be used for estimating the digestibility of pig diets and ii) determine whether this technique is more accurate with or without pre-treatment with NDS and α-amylase. This method is commonly used as an internal marker for the estimation of diet digestibility in ruminants. By its simplicity, this method is in theory interesting. The authors have used the data of a previous experiment with a control and a low crude protein diet, were total collection of faeces and urine has been performed in pigs. They then test the digestibility by the degradation of NDF in vitro by two method consisting of a pre-treatment or not. This is a first step validation as the authors said, since they only have 8 data in each group. Although fully relevant, I have some comments to improve the paper before scientific publication.
First, I recommend to improve the statistical analysis by presenting a graph of the raw data to see the overall link and the hierarchy of the point within dietary treatment and method (relationship between in vivo and pre-treated and in vivo and unpre-treated for CONV and LP) rather than looking only at the average. This means having intercept, slope, R2 and mean square prediction error (MSPE) for each comparison. This methodology will be important also if the authors test the impact of other type of diets (e.g. high fiber, high starch) to assess the overall accuracy of the method they propose.
Second, the discussion is lacking some elements on the effect of the type of NDF supply (e.g. changing the raw material) to have an idea about the possibility of extrapolating the method to different type of diet. As an example, the diet used in this experiment provided 12% NDF which is higher than corn and soybean meal-based diet commonly used in growing pigs in some contexts and which is lower than a high by-product diet used in finishing pigs that can go up to 17% NDF. Perhaps ruminant data can be used for that part.
Specific comments:
(Page/line)
1/12-14: please add details about in vivo vs in situ, what is in situ exactly?
1/21-22: As the authors concluded, and I agree, that further studies are needed to ensure the method works in a wider range of diet type, change can be for the method could be an alternative.
Author Response
Dear reviewer, here you can find our reply to your comments.
Thank you.

Reviewer 3 Report
Review of the manuscript No 939326 „Use of undigested NDF for estimation of diet digestibility in growing pigs”, Animals
The authors undertook an interesting work, aimed at developing a non-invasive in vitro technique for assessing the digestibility of nutrients in pig feed. In the modern approach to experimental animals, the use of alternative testing methods, which allowed to reduce the use of live animals for experiments, is desirable. However, the assumptions of their work are not clear. As is well known, modern pig nutrition is based on an accurate assessment of the digestibility of the nutrients of the feed, where the reference point is not the total faecal digestibility measured at the end of the large intestine, but the intestinal digestibility (up to the end of the small intestine). Thus, the comparison of in vitro and in vivo methods, should take into account the results of the intestinal digestibility not the total tract faecal digestibility. Moreover, digestible protein, or even more actually digestible amino acids, are considered the main nutrients for the balance of pig feeds. Not so much dry matter digestibility, or even more so, the NDFs which were studied by the authors - these two are rather typical of ruminant. In summary, the idea may be interesting, but I do not see this new method (DaisyII incubator) as a good choice, as well as the possibility of applying the obtained results to pig nutrition are doubtful. Moreover, the studying of only 3 parameters is rather modest for a scientific publication. In my opinion the manuscript should not be published.
- L 12: not „reduce the manure excretion”, but rather: reduce the amount of undigested nutrients excreted in the manure.
- L 15: there is „in vivo method with undigested neutral detergent fibre (uNDF)” and L 25:
„in vivo 24 method with uNDF” and L 142: „in vivo digestibility with uNDF„ and some more. But there is „In Vitro uNDF Determination” in the chapter Materials and Methods. So, in vivo or in vitro?
Author Response

(The authors gave the same response as above.)
